# Straggler Mitigation in Distributed Optimization Through Data Encoding

**Can Karakus**
UCLA
Los Angeles, CA
karakus@ucla.edu

**Yifan Sun**
Technicolor Research
Los Altos, CA
Yifan.Sun@technicolor.com

**Suhas Diggavi**
UCLA
Los Angeles, CA
suhasdiggavi@ucla.edu

**Wotao Yin**
UCLA
Los Angeles, CA
wotaoyin@math.ucla.edu

## Abstract

Slow running or straggler tasks can significantly reduce computation speed in distributed computation. Recently, coding-theory-inspired approaches have been applied to mitigate the effect of straggling, through embedding redundancy in certain linear computational steps of the optimization algorithm, thus completing the computation without waiting for the stragglers. In this paper, we propose an alternate approach where we embed the redundancy directly in the data itself, and allow the computation to proceed completely oblivious to encoding. We propose several encoding schemes, and demonstrate that popular batch algorithms, such as gradient descent and L-BFGS, applied in a coding-oblivious manner, deterministically achieve sample path linear convergence to an approximate solution of the original problem, using an arbitrarily varying subset of the nodes at each iteration. Moreover, this approximation can be controlled by the amount of redundancy and the number of nodes used in each iteration. We provide experimental results demonstrating the advantage of the approach over uncoded and data replication strategies.

## 1 Introduction

Solving large-scale optimization problems has become feasible through distributed implementations. However, the efficiency can be significantly hampered by slow processing nodes, network delays or node failures. In this paper we develop an optimization framework based on encoding the dataset, which mitigates the effect of straggler nodes in the distributed computing system. Our approach can be readily adapted to the existing distributed computing infrastructure and software frameworks, since the node computations are oblivious to the data encoding.

In this paper, we focus on problems of the form

$$\min_{w \in \mathbb{R}^p} f(w) := \frac{1}{2n} \min_{w \in \mathbb{R}^p} \|Xw - y\|^2, \tag{1}$$

where $X \in \mathbb{R}^{n \times p}$, $y \in \mathbb{R}^{n \times 1}$ represent the data matrix and vector respectively. The function $f(w)$ is mapped onto a distributed computing setup depicted in Figure 1, consisting of one central server and $m$ worker nodes, which collectively store the row-partitioned matrix $X$ and vector $y$. We focus on batch, synchronous optimization methods, where the delayed or failed nodes can significantly slow down the overall computation. Note that asynchronous methods are inherently robust to delays caused

by stragglers, although their convergence rates can be worse than their synchronous counterparts. Our approach consists of adding redundancy by encoding the data $X$ and $y$ into $\widetilde{X} = SX$ and $\widetilde{y} = Sy$, respectively, where $S \in \mathbb{R}^{(\beta n) \times n}$ is an encoding matrix with redundancy factor $\beta \geq 1$, and solving the effective problem

$$\min_{w \in \mathbb{R}^p} \widetilde{f}(w) := \min_{w \in \mathbb{R}^p} \frac{1}{2\beta n} \|S(Xw - y)\|^2 = \min_{w \in \mathbb{R}^p} \frac{1}{2\beta n} \|\widetilde{X}w - \widetilde{y}\|^2 \qquad (2)$$

instead. In doing so, we proceed with the computation in each iteration without waiting for the stragglers, with the idea that the inserted redundancy will compensate for the lost data. The goal is to design the matrix $S$ such that, when the nodes *obliviously* solve the problem (2) without waiting for the slowest $(m - k)$ nodes (where $k$ is a design parameter) the achieved solution approximates the original solution $w^* = \arg\min_w f(w)$ sufficiently closely. Since in large-scale machine learning and data analysis tasks one is typically not interested in the exact optimum, but rather a "sufficiently" good solution that achieves a good generalization error, such an approximation could be acceptable in many scenarios. Note also that the use of such a technique does not preclude the use of other, non-coding straggler-mitigation strategies (see [24] and references therein), which can still be implemented on top of the redundancy embedded in the system, to potentially further improve performance.

Focusing on gradient descent and L-BFGS algorithms, we show that under a spectral condition on $S$, one can achieve an approximation of the solution of (1), by solving (2), without waiting for the stragglers. We show that with sufficient redundancy embedded, and with updates from a sufficiently large, yet strict subset of the nodes in each iteration, it is possible to *deterministically* achieve linear convergence to a neighborhood of the solution, as opposed to convergence in expectation (see Fig. 4). Further, one can adjust the approximation guarantee by increasing the redundancy and number of node updates waited for in each iteration. Another potential advantage of this strategy is privacy, since the nodes do not have access to raw data itself, but can still perform the optimization task over the jumbled data to achieve an approximate solution.

Although in this paper we focus on quadratic objectives and two specific algorithms, in principle our approach can be generalized to more general, potentially non-smooth objectives and constrained optimization problems, as we discuss in Section 4 ( adding a regularization term is also a simple generalization).

Our main contributions are as follows. (i) We demonstrate that gradient descent (with constant step size) and L-BFGS (with line search) applied in a coding-oblivious manner on the encoded problem, achieves (universal) sample path linear convergence to an approximate solution of the original problem, using only a fraction of the nodes at each iteration. (ii) We present three classes of coding matrices; namely, equiangular tight frames (ETF), fast transforms, and random matrices, and discuss their properties. (iii) We provide experimental results demonstrating the advantage of the approach over uncoded $(S = I)$ and data replication strategies, for ridge regression using synthetic data on an AWS cluster, as well as matrix factorization for the Movielens 1-M recommendation task.

**Related work.** Use of data replication to aid with the straggler problem has been proposed and studied in [22, 1], and references therein. Additionally, use of coding in distributed computing has been explored in [13, 7]. However, these works exclusively focused on using coding at the computation level, *i.e.*, certain linear computational steps are performed in a coded manner, and explicit encoding/decoding operations are performed at each step. Specifically, [13] used MDS-coded distributed matrix multiplication and [7] focused on breaking up large dot products into shorter dot products, and perform redundant copies of the short dot products to provide resilience against stragglers. [21] considers a gradient descent method on an architecture where each data sample is replicated across nodes, and designs a code such that the exact gradient can be recovered as long as fewer than a certain number of nodes fail. However, in order to recover the exact gradient under any potential set of stragglers, the required redundancy factor is on the order of the number of straggling nodes, which could mean a large amount of overhead for a large-scale system. In contrast, we show that one can converge to an approximate solution with a redundancy factor independent of network size or problem dimensions (*e.g.,* 2 as in Section 5).

Our technique is also closely related to randomized linear algebra and sketching techniques [14, 6, 17], used for dimensionality reduction of large convex optimization problems. The main difference between this literature and the proposed coding technique is that the former focuses on reducing the problem dimensions to lighten the computational load, whereas coding *increases* the dimensionality

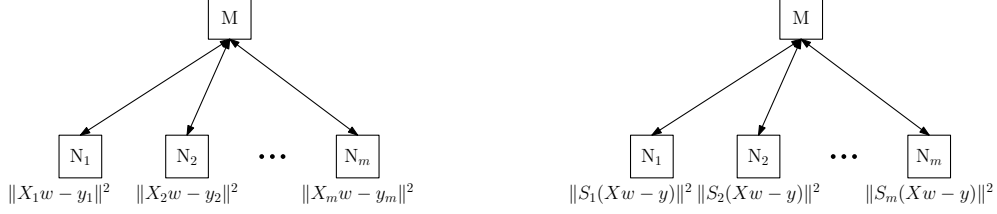

Figure 1: **Left:** Uncoded distributed optimization with partitioning, where $X$ and $y$ are partitioned as $X = \begin{bmatrix} X_1^\top & X_2^\top & \dots & X_m^\top \end{bmatrix}^\top$ and $y = \begin{bmatrix} y_1^\top & y_2^\top & \dots & y_m^\top \end{bmatrix}^\top$. **Right:** Encoded distributed optimization, where node $i$ stores $(S_i X, S_i y)$, instead of $(X_i, y_i)$. The uncoded case corresponds to $S = I$.

of the problem to provide robustness. As a result of the increased dimensions, coding can provide a much closer approximation to the original solution compared to sketching techniques.

A longer version of this paper is available on [12].

## 2   Encoded Optimization Framework

Figure 1 shows a typical data-distributed computational model in large-scale optimization (left), as well as our proposed encoded model (right). Our computing network consists of $m$ machines, where machine $i$ stores $\left(\widetilde{X}_i, \widetilde{y}_i\right) = (S_i X, S_i y)$ and $S = \begin{bmatrix} S_1^\top & S_2^\top & \dots & S_m^\top \end{bmatrix}^\top$. The optimization process is oblivious to the encoding, *i.e.*, once the data is stored at the nodes, the optimization algorithm proceeds exactly as if the nodes contained uncoded, raw data $(X, y)$. In each iteration $t$, the central server broadcasts the current estimate $w_t$, and each worker machine computes and sends to the server the gradient terms corresponding to its own partition $g_i(w_t) := \widetilde{X}_i^\top(\widetilde{X}_i w_t - \widetilde{y}_i)$.

Note that this framework of distributed optimization is typically communication-bound, where communication over a few slow links constitute a significant portion of the overall computation time. We consider a strategy where at each iteration $t$, the server only uses the gradient updates from the first $k$ nodes to respond in that iteration, thereby preventing such slow links and straggler nodes from stalling the overall computation:

$$\widetilde{g}_t = \frac{1}{2\beta\eta n} \sum_{i \in A_t} g_i(w_t) = \frac{1}{\beta\eta n} \widetilde{X}_A^\top(\widetilde{X}_A w_t - \widetilde{y}_A),$$

where $A_t \subseteq [m]$, $|A_t| = k$ are the indices of the first $k$ nodes to respond at iteration $t$, $\eta := \frac{k}{m}$ and $\widetilde{X}_A = [S_i X]_{i \in A_t}$. (Similarly, $S_A = [S_i]_{i \in A_t}$.) Given the gradient approximation, the central server then computes a *descent direction* $d_t$ through the history of gradients and parameter estimates. For the remaining nodes $i \notin A_t$, the server can either send an interrupt signal, or simply drop their updates upon arrival, depending on the implementation.

Next, the central server chooses a step size $\alpha_t$, which can be chosen as constant, decaying, or through exact line search [1] by having the workers compute $\widetilde{X}d_t$ that is needed to compute the step size. We again assume the central server only hears from the fastest $k$ nodes, denoted by $D_t \subseteq [m]$, where $D_t \neq A_t$ in general, to compute

$$\alpha_t = -\nu \frac{d_t^\top \widetilde{g}_t}{d_t^\top \widetilde{X}_D^\top \widetilde{X}_D d_t}, \tag{3}$$

where $\widetilde{X}_D = [S_i X]_{i \in D_t}$, and $0 < \nu < 1$ is a back-off factor of choice.

Our goal is to especially focus on the case $k < m$, and design an encoding matrix $S$ such that, for any sequence of sets $\{A_t\}, \{D_t\}$, $f(w_t)$ universally converges to a neighborhood of $f(w^*)$. Note that in general, this scheme with $k < m$ is not guaranteed to converge for traditionally batch methods like L-BFGS. Additionally, although the algorithm only works with the encoded function $\widetilde{f}$, our goal is to provide a convergence guarantee in terms of the *original* function $f$.

## 3 Algorithms and Convergence Analysis

Let the smallest and largest eigenvalues of $X^\top X$ be denoted by $\mu > 0$ and $M > 0$, respectively.

Let $\eta$ with $\frac{1}{\beta} < \eta \leq 1$ be given. In order to prove convergence, we will consider a family of matrices $\{S^{(\beta)}\}$ where $\beta$ is the aspect ratio (redundancy factor), such that for any $\epsilon > 0$, and any $A \subseteq [m]$ with $|A| = \eta m$,

$$(1 - \epsilon)I \preceq S_A^\top S_A \preceq (1 + \epsilon)I, \tag{4}$$

for sufficiently large $\beta \geq 1$, where $S_A = [S_i]_{i \in A}$ is the submatrix associated with subset $A$ (we drop dependence on $\beta$ for brevity). Note that this is similar to the restricted isometry property (RIP) used in compressed sensing [4], except that (4) is only required for submatrices of the form $S_A$. Although this condition is needed to prove worst-case convergence results, in practice the proposed encoding scheme can work well even when it is not exactly satisfied, as long as the bulk of the eigenvalues of $S_A^\top S_A$ lie within a small interval $[1 - \epsilon, 1 + \epsilon]$. We will discuss several specific constructions and their relation to property (4) in Section 4.

**Gradient descent.** We consider gradient descent with constant step size, *i.e.*,

$$w_{t+1} = w_t + \alpha d_t = w_t - \alpha \widetilde{g}_t.$$

The following theorem characterizes the convergence of the encoded problem under this algorithm.

**Theorem 1.** *Let $f_t = f(w_t)$, where $w_t$ is computed using gradient descent with updates from a set of (fastest) workers $A_t$, with constant step size $\alpha_t \equiv \alpha = \frac{2\zeta}{M(1+\epsilon)}$ for some $0 < \zeta \leq 1$, for all t. If S satisfies (4) with $\epsilon > 0$, then for all sequences of $\{A_t\}$ with cardinality $|A_t| = k$,*

$$f_t \leq (\kappa\gamma_1)^t f_0 + \frac{\kappa^2(\kappa - \gamma_1)}{1 - \kappa\gamma_1} f(w^*), \quad t = 1, 2, \ldots,$$

*where $\kappa = \frac{1+\epsilon}{1-\epsilon}$, and $\gamma_1 = \left(1 - \frac{4\mu\zeta(1-\zeta)}{M(1+\epsilon)}\right)$, and $f_0 = f(w_0)$ is the initial objective value.*

The proof is provided in Appendix B of [12], which relies on the fact that the solution to the effective "instantaneous" problem corresponding to the subset $A_t$ lies in the set $\{w : f(w) \leq \kappa^2 f(w^*)\}$, and therefore each gradient descent step attracts the estimate towards a point in this set, which must eventually converge to this set. Note that in order to guarantee linear convergence, we need $\kappa\gamma_1 < 1$, which can be ensured by property (4).

Theorem 1 shows that gradient descent over the encoded problem, based on updates from only $k < m$ nodes, results in *deterministically* linear convergence to a neighborhood of the true solution $w^*$, for sufficiently large $k$, as opposed to convergence in expectation. Note that by property (4), by controlling the redundancy factor $\beta$ and the number of nodes $k$ waited for in each iteration, one can control the approximation guarantee. For $k = m$ and $S$ designed properly (see Section 4), then $\kappa = 1$ and the optimum value of the original function $f(w^*)$ is reached.

**Limited-memory-BFGS.** Although L-BFGS is originally a batch method, requiring updates from all nodes, its stochastic variants have also been proposed recently [15, 3]. The key modification to ensure convergence is that the Hessian estimate must be computed via gradient components that are common in two consecutive iterations, *i.e.*, from the nodes in $A_t \cap A_{t-1}$. We adapt this technique to our scenario. For $t > 0$, define $u_t := w_t - w_{t-1}$, and

$$r_t := \frac{m}{2\beta n |A_t \cap A_{t-1}|} \sum_{i \in A_t \cap A_{t-1}} (g_i(w_t) - g_i(w_{t-1})).$$

Then once the gradient terms $\{g_t\}_{i \in A_t}$ are collected, the descent direction is computed by $d_t = -B_t \widetilde{g}_t$, where $B_t$ is the inverse Hessian estimate for iteration $t$, which is computed by

$$B_t^{(\ell+1)} = V_{j_\ell}^\top B_t^{(\ell)} V_{j_\ell} + \rho_{j_\ell} u_{j_\ell} u_{j_\ell}^\top, \quad \rho_k = \frac{1}{r_k^\top u_k}, \quad V_k = I - \rho_k r_k u_k^\top$$

with $j_\ell = t - \widetilde{\sigma} + \ell$, $B_t^{(0)} = \frac{r_t^\top r_t}{r_t^\top u_t} I$, and $B_t := B_t^{(\widetilde{\sigma})}$ with $\widetilde{\sigma} := \min\{t, \sigma\}$, where $\sigma$ is the L-BFGS memory length. Once the descent direction $d_t$ is computed, the step size is determined through exact line search, using (3), with back-off factor $\nu = \frac{1-\epsilon}{1+\epsilon}$, where $\epsilon$ is as in (4).

For our convergence result for L-BFGS, we need another assumption on the matrix $S$, in addition to (4). Defining $\check{S}_t = [S_i]_{i \in A_t \cap A_{t-1}}$ for $t > 0$, we assume that for some $\delta > 0$,

$$\delta I \preceq \check{S}_t^\top \check{S}_t \tag{5}$$

for all $t > 0$. Note that this requires that one should wait for sufficiently many nodes to finish so that the overlap set $A_t \cap A_{t-1}$ has more than a fraction $\frac{1}{\beta}$ of all nodes, and thus the matrix $\check{S}_t$ can be full rank. This is satisfied if $\eta \geq \frac{1}{2} + \frac{1}{2\beta}$ in the worst-case, and under the assumption that node delays are i.i.d., it is satisfied in expectation if $\eta \geq \frac{1}{\sqrt{\beta}}$. However, this condition is only required for a worst-case analysis, and the algorithm may perform well in practice even when this condition is not satisfied. The following lemma shows the stability of the Hessian estimate.

**Lemma 1.** *If* (5) *is satisfied, then there exist constants* $c_1, c_2 > 0$ *such that for all* $t$, *the inverse Hessian estimate* $B_t$ *satisfies* $c_1 I \preceq B_t \preceq c_2 I$.

The proof, provided in Appendix A of [12], is based on the well-known trace-determinant method. Using Lemma 1, we can show the following result.

**Theorem 2.** *Let* $f_t = f(w_t)$, *where* $w_t$ *is computed using L-BFGS as described above, with gradient updates from machines* $A_t$, *and line search updates from machines* $D_t$. *If* $S$ *satisfies* (4) *and* (5), *for all sequences of* $\{A_t\}, \{D_t\}$ *with* $|A_t| = |D_t| = k$,

$$f_t \leq (\kappa \gamma_2)^t f_0 + \frac{\kappa^2(\kappa - \gamma_2)}{1 - \kappa \gamma_2} f(w^*),$$

*where* $\kappa = \frac{1+\epsilon}{1-\epsilon}$, *and* $\gamma_2 = \left(1 - \frac{4\mu c_1 c_2}{M(c_1+c_2)^2}\right)$, *and* $f_0 = f(w_0)$ *is the initial objective value.*

The proof is provided in Appendix B of [12]. Similar to Theorem 1, the proof is based on the observation that the solution of the effective problem at time $t$ lies in a bounded set around the true solution $w^*$. As in gradient descent, coding enables linear convergence deterministically, unlike the stochastic and multi-batch variants of L-BFGS [15, 3].

**Generalizations.** Although we focus on quadratic cost functions and two specific algorithms, our approach can potentially be generalized for objectives of the form $\|Xw - y\|^2 + h(w)$ for a simple convex function $h$, *e.g.*, LASSO; or constrained optimization $\min_{w \in C} \|Xw - y\|^2$ (see [11]); as well as other first-order algorithms used for such problems, *e.g.*, FISTA [2]. In the next section we demonstrate that the codes we consider have desirable properties that readily extend to such scenarios.

## 4 Code Design

We consider three classes of coding matrices: tight frames, fast transforms, and random matrices.

**Tight frames.** A unit-norm *frame* for $\mathbb{R}^n$ is a set of vectors $F = \{\phi_i\}_{i=1}^{n\beta}$ with $\|\phi_i\| = 1$, where $\beta \geq 1$, such that there exist constants $\xi_1 \geq \xi_2 > 0$ such that, for any $u \in \mathbb{R}^n$,

$$\xi_1 \|u\|^2 \leq \sum_{i=1}^{n\beta} |\langle u, \phi_i \rangle|^2 \leq \xi_2 \|u\|^2.$$

The frame is *tight* if the above satisfied with $\xi_1 = \xi_2$. In this case, it can be shown that the constants are equal to the redundancy factor of the frame, *i.e.*, $\xi_1 = \xi_2 = \beta$. If we form $S \in \mathbb{R}^{(\beta n) \times n}$ by rows that are a *tight frame*, then we have $S^\top S = \beta I$, which ensures $\|Xw - y\|^2 = \frac{1}{\beta} \|SXw - Sy\|^2$. Then for any solution $\widetilde{w}^*$ to the encoded problem (with $k = m$),

$$\nabla \widetilde{f}(\widetilde{w}^*) = X^\top S^\top S(X\widetilde{w}^* - y) = \beta(X\widetilde{w}^* - y)^\top X = \beta \nabla f(\widetilde{w}^*).$$

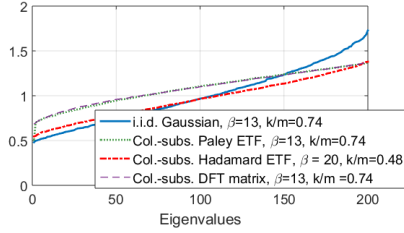

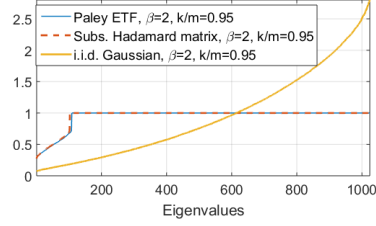

Figure 2: Sample spectrum of $S_A^\top S_A$ for various constructions with high redundancy, and relatively small $k$ (normalized).

Figure 3: Sample spectrum of $S_A^\top S_A$ for various constructions with low redundancy, and large $k$ (normalized).

Therefore, the solution to the encoded problem satisfies the optimality condition for the original problem as well:

$$\nabla \widetilde{f}(\widetilde{w}^*) = 0, \quad \Leftrightarrow \quad \nabla f(\widetilde{w}^*) = 0,$$

and if $f$ is also strongly convex, then $\widetilde{w}^* = w^*$ is the unique solution. Note that since the computation is coding-oblivious, this is not true in general for an arbitrary full rank matrix, and this is, in addition to property (4), a desired property of the encoding matrix. In fact, this equivalency extends beyond smooth unconstrained optimization, in that

$$\left\langle \nabla \widetilde{f}(\widetilde{w}^*), w - \widetilde{w}^* \right\rangle \geq 0, \ \forall w \in \mathcal{C} \quad \Leftrightarrow \quad \langle \nabla f(\widetilde{w}^*), w - \widetilde{w}^* \rangle \geq 0, \ \forall w \in \mathcal{C}$$

for any convex constraint set $\mathcal{C}$, as well as

$$-\nabla \widetilde{f}(\widetilde{w}^*) \in \partial h(\widetilde{w}^*), \quad \Leftrightarrow \quad -\nabla f(\widetilde{w}^*) \in \partial h(\widetilde{w}^*),$$

for any non-smooth convex objective term $h(x)$, where $\partial h$ is the subdifferential of $h$. This means that tight frames can be promising encoding matrix candidates for non-smooth and constrained optimization too. In [11], it was shown that when $\{A_t\}$ is static, equiangular tight frames allow for a close approximation of the solution for constrained problems.

A tight frame is *equiangular* if $|\langle \phi_i, \phi_j \rangle|$ is constant across all pairs $(i, j)$ with $i \neq j$.

**Proposition 1** (Welch bound [23]). *Let $F = \{\phi_i\}_{i=1}^{n\beta}$ be a tight frame. Then $\omega(F) \geq \sqrt{\frac{\beta-1}{2n\beta-1}}$. Moreover, equality is satisfied if and only if $F$ is an equiangular tight frame.*

Therefore, an ETF minimizes the correlation between its individual elements, making each submatrix $S_A^\top S_A$ as close to orthogonal as possible, which is promising in light of property (4). We specifically evaluate Paley [16, 10] and Hadamard ETFs [20] (not to be confused with Hadamard matrix, which is discussed next) in our experiments. We also discuss Steiner ETFs [8] in Appendix D of [12], which enable efficient implementation.

**Fast transforms.** Another computationally efficient method for encoding is to use fast transforms: Fast Fourier Transform (FFT), if $S$ is chosen as a subsampled DFT matrix, and the Fast Walsh-Hadamard Transform (FWHT), if $S$ is chosen as a subsampled real Hadamard matrix. In particular, one can insert rows of zeroes at random locations into the data pair $(X, y)$, and then take the FFT or FWHT of each column of the augmented matrix. This is equivalent to a randomized Fourier or Hadamard ensemble, which is known to satisfy the RIP with high probability [5].

**Random matrices.** A natural choice of encoding is using i.i.d. random matrices. Although such random matrices do not have the computational advantages of fast transforms or the optimality-preservation property of tight frames, their eigenvalue behavior can be characterized analytically. In particular, using the existing results on the eigenvalue scaling of large i.i.d. Gaussian matrices [9, 19] and union bound, it can be shown that

$$\mathbb{P}\left(\max_{A:|A|=k} \lambda_{\max}\left(\frac{1}{\beta \eta n} S_A^\top S_A\right) > \left(1 + \sqrt{\frac{1}{\beta \eta}}\right)^2\right) \to 0, \tag{6}$$

$$\mathbb{P}\left(\min_{A:|A|=k} \lambda_{\min}\left(\frac{1}{\beta \eta n} S_A^\top S_A\right) < \left(1 - \sqrt{\frac{1}{\beta \eta}}\right)^2\right) \to 0, \tag{7}$$

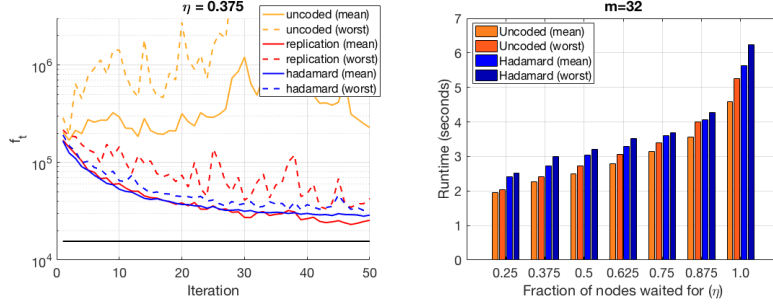

Figure 4: **Left:** Sample evolution of uncoded, replication, and Hadamard (FWHT)-coded cases, for $k = 12$, $m = 32$. **Right:** Runtimes of the schemes for different values of $\eta$, for the same number of iterations for each scheme. Note that this essentially captures the delay profile of the network, and does not reflect the relative convergence rates of different methods.

as $n \to \infty$, where $\sigma_i$ denotes the $i$th singular value. Hence, for sufficiently large redundancy and problem dimension, i.i.d. random matrices are good candidates for encoding as well. However, for finite $\beta$, even if $k = m$, in general for this encoding scheme the optimum of the original problem is not recovered exactly.

**Property** (4) **and redundancy requirements.** Using the analytical bounds (6)–(7) on i.i.d. Gaussian matrices, one can see that such matrices satisfy (4) with $\epsilon = O\left(\frac{1}{\sqrt{\beta \eta}}\right)$, independent of problem dimensions or number of nodes $m$. Although we do not have tight eigenvalue bounds for subsampled ETFs, numerical evidence (Figure 2) suggests that they may satisfy (4) with smaller $\epsilon$ than random matrices, and thus we believe that the required redundancy in practice is even smaller for ETFs.

Note that our theoretical results focus on the extreme eigenvalues due to a worst-case analysis; in practice, most of the energy of the gradient will be on the eigen-space associated with the bulk of the eigenvalues, which the following proposition suggests can be mostly 1 (also see Figure 3), which means even if (4) is not satisfied, the gradient (and the solution) can be approximated closely for a modest redundancy, such as $\beta = 2$. The following result is a consequence of the Cauchy interlacing theorem, and the definition of tight frames.

**Proposition 2.** *If the rows of $S$ are chosen to form an ETF with redundancy $\beta$, then for $\eta \geq 1 - \frac{1}{\beta}$, $\frac{1}{\beta} S_A^\top S_A$ has $n(1 - \beta\eta)$ eigenvalues equal to 1.*

## 5 Numerical Results

**Ridge regression with synthetic data on AWS EC2 cluster.** We generate the elements of matrix $X$ i.i.d. $\sim N(0,1)$, the elements of $y$ i.i.d. $\sim N(0,p)$, for dimensions $(n,p) = (4096, 6000)$, and solve the problem $\min_w \frac{1}{2\beta n} \left\| \widetilde{X}w - \widetilde{y} \right\|^2 + \frac{\lambda}{2} \|w\|^2$, for regularization parameter $\lambda = 0.05$. We evaluate column-subsampled Hadamard matrix with redundancy $\beta = 2$ (encoded using FWHT for fast encoding), data replication with $\beta = 2$, and uncoded schemes. We implement distributed L-BFGS as described in Section 3 on an Amazon EC2 cluster using the `mpi4py` Python package, over $m = 32$ `m1.small` worker node instances, and a single `c3.8xlarge` central server instance. We assume the central server encodes and sends the data variables to the worker nodes (see Appendix D of [12] for a discussion of how to implement this more efficiently).

Figure 4 shows the result of our experiments, which are aggregated over 20 trials. As baselines, we consider the uncoded scheme, as well as a replication scheme, where each uncoded partition is replicated $\beta = 2$ times across nodes, and the server uses the faster copy in each iteration. It can be seen from the right figure that one can speed up computation by reducing $\eta$ from 1 to, for instance, 0.375, resulting in more than $40\%$ reduction in the runtime. Note that in this case, uncoded L-BFGS fails to converge, whereas the Hadamard-coded case stably converges. We also observe that the data replication scheme converges on average, but in the worst case, the convergence is much less smooth, since the performance may deteriorate if both copies of a partition are delayed.

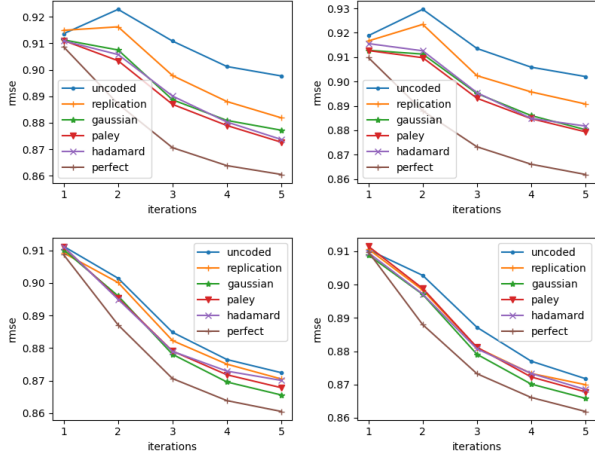

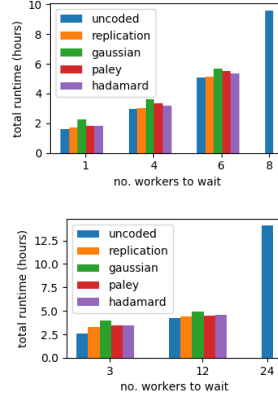

Figure 5: Test RMSE for $m = 8$ (left) and $m = 24$ (right) nodes, where the server waits for $k = m/8$ (top) and $k = m/2$ (bottom) responses. "Perfect" refers to the case where $k = m$.

Figure 6: Total runtime with $m = 8$ and $m = 24$ nodes for different values of $k$, under fixed 100 iterations for each scheme.

**Matrix factorization on Movielens 1-M dataset.** We next apply matrix factorization on the MovieLens-1M dataset [18] for the movie recommendation task. We are given $R$, a sparse matrix of movie ratings 1–5, of dimension $\#users \times \#movies$, where $R_{ij}$ is specified if user $i$ has rated movie $j$. We withhold randomly 20% of these ratings to form an 80/20 train/test split. The goal is to recover user vectors $x_i \in \mathbb{R}^p$ and movie vectors $y_i \in \mathbb{R}^p$ (where $p$ is the embedding dimension) such that $R_{ij} \approx x_i^T y_j + u_i + v_j + \mu$, where $u_i$, $v_j$, and $\mu$ are user, movie, and global biases, respectively. The optimization problem is given by

$$\min_{x_i, y_j, u_i, v_j} \sum_{i,j: \text{ observed}} (R_{ij} - u_i - v_j - x_i^T y_j - \mu)^2 + \lambda \left( \sum_i \|x_i\|_2^2 + \|u\|_2^2 + \sum_j \|y_j\|_2^2 + \|v\|_2^2 \right). \tag{8}$$

We choose $\mu = 3$, $p = 15$, and $\lambda = 10$, which achieves a test RMSE 0.861, close to the current best test RMSE on this dataset using matrix factorization[2].

Problem (8) is often solved using alternating minimization, minimizing first over all $(x_i, u_i)$, and then all $(y_j, v_j)$, in repetition. Each such step further decomposes by row and column, made smaller by the sparsity of $R$. To solve for $(x_i, u_i)$, we first extract $I_i = \{j \mid r_{ij} \text{ is observed}\}$, and solve the resulting sequence of regularized least squares problems in the variables $w_i = [x_i^\top, u_i]^\top$ distributedly using coded L-BFGS; and repeat for $w = [y_j^\top, v_j]^\top$, for all $j$. As in the first experiment, distributed coded L-BFGS is solved by having the master node encoding the data locally, and distributing the encoded data to the worker nodes (Appendix D of [12] discusses how to implement this step more efficiently). The overhead associated with this initial step is included in the overall runtime in Figure 6.

The Movielens experiment is run on a single 32-core machine with 256 GB RAM. In order to simulate network latency, an artificial delay of $\Delta \sim \exp(10 \text{ ms})$ is imposed each time the worker completes a task. Small problem instances ($n < 500$) are solved locally at the central server, using the built-in function `numpy.linalg.solve`. Additionally, parallelization is only done for the ridge regression instances, in order to isolate speedup gains in the L-BFGS distribution. To reduce overhead, we create a bank of encoding matrices $\{S_n\}$ for Paley ETF and Hadamard ETF, for $n = 100, 200, \ldots, 3500$, and then given a problem instance, subsample the columns of the appropriate matrix $S_n$ to match the dimensions. Overall, we observe that encoding overhead is amortized by the speed-up of the distributed optimization.

Figure 5 gives the final performance of our distributed L-BFGS for various encoding schemes, for each of the 5 epochs, which shows that coded schemes are most robust for small $k$. A full table of results is given in Appendix C of [12].

**Acknowledgments**

This work was supported in part by NSF grants 1314937 and 1423271.

## Footnotes

[1]Note that exact line search is not more expensive than backtracking line search for a quadratic loss, since it only requires a single matrix-vector multiplication.

[2]http://www.mymedialite.net/examples/datasets.html

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
