[Reviews · NeurIPS 2017]

Reviewer 1



The authors propose a simple scheme for making the distributed solution of large least squares problems more predictable by avoid tail events such as stragglers, slowing down the computation. While the setup is not as general as that of [19], the results are much stronger. Empirically the data increases only a factor of 2 while allowing a great amount of leeway on how many machines they are going to wait for in each iteration and still achieve a good final solution. The authors failed to make connections with Randomized Linear Algebra techniques for solving least squares problems. In particular it is known that to guarrantee a good solution for least squares with sampling data points should be sampled according to their leverage score. One option proposed in the RandLA community is to apply subsampled Hadamard transforms because they uniformize the leverage scores, making the subsequent selection of transformed data points, trivial. I think the authors here might be observing some of the benefits of this. To verify, it would be good to perform an experiment with a datasets where the data points have very non-uniform leverage scores.

Reviewer 2



This paper addresses the issue of performing distributed optimization in the presence of straggling/slow computation units. In particular, the paper focuses on the problem of linear regression min_w \|Xw - y\|_2, ---- (1) where X = [(X_1)^T, (X_2)^T,..., (X_m)^T]^T and y = [y_1, y_2,..., y_m]^T denote the data points and the corresponding labels. In general, the distributed setup with $m$ worker nodes allocates $i$-th data point $X_i$ and the associated label $y_i$ to $i$-th worker node. The linear regression problem is then solved in an iterative manner where messages/information needs to be communicated among (master) server and the worker nodes. However, in practice, some of the workers (aka stragglers) take longer time to completer their end of processing/communication, which slows down the entire distributed optimization problem. In order to address the delay caused by the stragglers, this paper proposes to introduce redundancy into the data samples and labels by various encoding methods. The paper utilizes an encoding matrix $S$ to transforms the data points $X$ and labels $y$ into $\tilde{X} = SX$ and $\tilde{y} = Sy$, respectively. The paper then solves the following problem instead of (1). min_w \|\tilde{X}w - \tilde{y}\|_2 = min_w \|SXw - Sy}\|_2, where the optimization setup allocates the $i$-th row of the matrices $\tilde{X}$ and $\tilde{y}$ to the $i$-worker node. During each iteration of the distributed optimization process, the server only waits for a subset of worker nodes to respond and then proceeds with the next iteration of the optimization process based on the received information. In this way, the server ignores the stragglers. Interestingly, the paper shows that the introduced redundancy allows the server to compute an (almost) optimal solution of the original problem as long as the encoding matrix is chosen to satisfy certain spectral properties and the server does not discard too many worker nodes in each iteration. The paper focuses on two particular algorithms for solving the underlying optimization problem: 1) gradient descent and 2) L-BFGS. The paper shows that for suitable encoding matrices the proposed solution provably obtains an almost optimal solution. The paper then presents three different approached to generate the encoding matrices with the desired properties. The authors also present extensive simulation results to corroborate their theoretical findings. The paper addresses a very interesting and practical problem. The proposed solutions is interesting which builds on a recent trend of using coding ideas to mitigate stragglers during distributed computation. That said, the approaches adopted in the paper and its focus significantly differs from some of the existing papers in this area. The paper is also very well written and addresses the underlying problem in a comprehensive manner. I have a minor comment for the authors. While discussing the related work, the authors argue that most of the previous works preform redundant computation whereas this paper directly encodes the data. However, the authors do acknowledge that [19] does perform a very simple form of data encoding (i.e., replication). The reviewer would like to point out that short-dot approach in [5] is similar to that in [19], therefore [5] can also be classified as encoding the data. In general, all the previous work including [8] can be viewed as encoding the data (assuming that, in a matrix vector computation, matrix is also part of data). Adopting this general view might be beneficial to develop a unified framework to address the problems in this domain.